# *Aspergillus* Section *Terrei* and Antifungals: From Broth to Agar-Based Susceptibility Testing Methods

**DOI:** 10.3390/jof9030306

**Published:** 2023-02-27

**Authors:** Roya Vahedi-Shahandashti, Lisa Hahn, Jos Houbraken, Cornelia Lass-Flörl

**Affiliations:** 1Institute of Hygiene and Medical Microbiology, Medical University of Innsbruck, 6020 Innsbruck, Austria; 2Westerdijk Fungal Biodiversity Institute, 3584 CT Utrecht, The Netherlands

**Keywords:** antifungal susceptibility testing, CLSI, EUCAST, epidemiological cutoff values, clinical breakpoints, antifungal resistance, *Aspergillus terreus*, amphotericin B, isavuconazole

## Abstract

Providing timely antifungal treatment to patients suffering from life-threatening invasive fungal infections (IFIs) is essential. Due to the changing epidemiology and the emergence of antifungal resistance in *Aspergillus*, the most commonly responsible mold of IFIs, antifungal susceptibility testing (AFST) has become increasingly important to guide clinical decisions. This study assessed the essential agreement (EA) between broth microdilution methods (the Clinical and Laboratory Standards Institute (CLSI) and the European Committee on Antimicrobial Susceptibility Testing (EUCAST)) and the Etest of amphotericin B (AmB), liposomal amphotericin B (L-AmB), and isavuconazole (ISA) against 112 *Aspergillus* section *Terrei*. An EA within ±2 dilutions of ≥90% between the two methods was considered acceptable. Excellent EA was found between EUCAST and CLSI of AmB and ISA (98.2% and 95.5%, respectively). The correlation of Etest results and EUCAST/CLSI was not acceptable (<90%) for any tested antifungal; however, Etest and CLSI for AmB (79.6%) and ISA (77.6%) showed a higher EA than Etest and EUCAST for AmB (49.5%) and ISA (46.4%). It was concluded that the Etest method requires its own clinical breakpoints (CBPs) and epidemiological cutoff values (ECVs), and interpreting Etest results using EUCAST and CLSI-adapted CBPs and ECVs could result in misinterpretation as Etest shows lower minimum inhibitory concentrations (MICs).

## 1. Introduction

Invasive fungal infections (IFIs) are becoming more prevalent due to increasing immunosuppressive drugs and immune-modulating diseases [1,2]. In severely immunocompromised patients, filamentous fungi are responsible for about 50% to 75% of IFIs, which are associated with high mortality and morbidity [1]. Invasive aspergillosis (IA) is the most reported mold infection in IFIs, most often caused by *Aspergillus fumigatus*, followed by other *Aspergillus* species, such as *A. terreus* [3,4]. Compared to other Aspergilli, A. terreus has an exceptional clinical status due to its high dissemination potential and reduced sensitivity to amphotericin B (AmB), the drug of choice for many severe fungal infections in vulnerable hosts [5,6,7,8]. *A. terreus* isolates exhibit a broad range of AmB minimum inhibitory concentration (MICs), from infrequently low MICs (≤1 mg/L) to frequently high MICs (≥2 mg/L), making clinical breakpoints (CBPs) difficult to establish [7,9,10,11,12]. The limited classes of available therapeutic antifungal agents and the rising number of antifungal resistance make the treatment of IFIs challenging [13,14,15]. Voriconazole remains the preferred agent for the treatment of aspergillosis, including *A. terreus* [16]. Alternative therapeutic options include isavuconazole, voriconazole plus an echinocandin, and liposomal AmB (L-AmB) [17]. As azole-resistance rates in *A. terreus* have been rising, other alternative therapeutic agents are being considered for investigation in the present study, including isavuconazole, another broad-spectrum azole, and conventional AmB and L-AmB, with less toxicity [14,17,18].

Selecting an antifungal agent for therapy depends on the fungal pathogen’s susceptibility to the antimycotic agent. Therefore, antifungal susceptibility testing (AFST) is becoming increasingly important for managing patient outcomes [19]. Determining the MIC using AFST provides an in vitro measure of the susceptibility of the causal agent [19]. Several AFST methods are currently used or under development. An ideal susceptibility testing method must be easy, reproducible, accurate, and cost-effective. In vitro AFST is influenced by many factors [20], so minimizing their influence on the final MIC value was the main reason behind standardization. The Clinical and Laboratory Standards Institute (CLSI) [21] and the European Committee on Antimicrobial Susceptibility Testing (EUCAST) [22] established and standardized two independent standard broth microdilution methods, frequently referred to as reference guidelines. Agar-based commercialized gradient diffusion strips (Etest) are the convenient approach for AFST in clinical practice compared to broth microdilution AFSTs, which are labor-intensive and time-consuming [23]. The best susceptibility testing method for molds is unclear. Therefore, the present study aimed to compare the agreement between Etest and two broth microdilution methods (CLSI and EUCAST) for AmB, L-AmB, and isavuconazole (ISA) against a collection of 112 *Aspergillus* section *Terrei* isolates.

## 2. Materials and Methods

### 2.1. Fungal Strains

A total of 112 molecularly identified *Aspergillus* section *Terrei* isolates, including *A. terreus* sensu stricto (s.s.) (n = 50), *A. hortai* (n = 11), *A. citrinoterreus* (n = 34), *A. alabamensis* (n = 9), *A. iranicus* (n = 5), *A. niveus* (n = 2), and *A. neoafricanus* (n = 1) were analyzed. The isolate collection included strains previously obtained and included in the ISHAM-ECMM-EFISG TerrNet Study (www.isham.org/working-groups/aspergillus-terreus (accessed on 24 February 2017)) [14,24] and those preserved in the CBS biobank housed at the Westerdijk Fungal Biodiversity Institute, Utrecht, the Netherlands. TerrNet Study [14] and CBS isolates [25] were identified according to the previous description.

### 2.2. Antifungal Agents

For AFST by broth microdilution, antifungal powders of deoxycholate AmB (Sigma-Aldrich, Vienna, Austria, A2411) (solvent; dimethyl sulfoxide, Sigma-Aldrich, Vienna, Austria), L-AmB (Gilead Sciences, Inc., Vienna, Austria, 020122D) (solvent; distilled water), and ISA (Sigma-Aldrich, Vienna, Austria, SML 000013488) (solvent; dimethyl sulfoxide, Sigma-Aldrich, Vienna, Austria) were utilized. For Etest AFST, commercialized gradient strips for AmB (0.002–32 mg/L; BioMérieux, Vienna, Austria) and ISA (0.002–32 mg/L; Liofilchem, Roseto degli Abruzzi, Italy) were applied.

### 2.3. Inoculum Preparation and AFST

Broth microdilution AFSTs were carried out according to CLSI [21] and EUCAST [22] guidelines. The Etest AFSTs for AmB and ISA were performed according to the instructions provided by the manufacturer. Etest strips of L-AmB were not commercially available and therefore not included in this study. Isolates were cultured from 10% glycerol frozen stocks (−80 °C) on malt extract agar (MEA) (Carl Roth, Karlsruhe, Germany) at 37 °C for up to 5 days; spores were harvested by applying spore suspension buffer (0.9% NaCl, 0.01% Tween 20 (Sigma-Aldrich, Vienna, Austria, P1379). Briefly, 90 mm diameter plates containing RPMI 1640 medium agar (Sigma-Aldrich, Vienna, Austria, R6504) supplemented with 2% glucose buffered to pH 7.0 with 0.165 M morpholinepropanesulfonic acid (MOPS) (Sigma-Aldrich, Vienna, Austria) were used [23,26]. The agar surface was inoculated with a swab dipped in a cell suspension adjusted to a turbidity of 0.5 McFarland standard (equivalent to 1 × 10^6^ CFU/mL). After 15 min, the strips were placed on the agar surface, and the plates were incubated at 37 °C. Etest MIC readings were taken after 24 h, except for strains with slow or insufficient growth, which were allowed an additional 24 h at room temperature, thus preventing overgrowth and making reading the results easier.

### 2.4. Interpretation of Results

A final reading of MIC results for broth microdilutions was performed with an inverted magnifying mirror after 48 h as the lowest drug concentration with complete inhibition of growth. MIC_50_ and MIC_90_ represent the MIC values at which 50% and 90% of the isolates in a test population are inhibited. The E-test MIC was the lowest drug concentration where the edge of the elliptical inhibition reached the antifungal strip’s scale. Etest MICs were rounded up to the next higher log2 dilution for comparison with broth microdilutions. Essential agreement (EA) between Etest and broth microdilution results was considered when the MIC values obtained with the methods fell within ±2 dilutions of the 2-fold dilution scheme [26,27]. EA values of ≥90% were regarded as acceptable.

## 3. Results

The MIC distribution and in vitro susceptibility testing results of 112 *Aspergillus* section *Terrei* isolates against AmB, L-AmB, and ISA performed by EUCAST, CLSI, and Etest methods are shown in Figure 1 and Table 1. The broth microdilution and Etest methods generated different MIC values. Considering all species, Etest MIC values for AmB and ISA were lower than broth microdilution values. MIC_90_ values of AmB and ISA obtained by CLSI were one-fold lower than EUCAST. Although MIC_90_ values of AmB gained by Etest and CLSI did not differ, MIC_90_ of ISA was different by two-fold. The MIC range of AmB was wider when tested by Etest (0.032–16 mg/L) than by CLSI and EUCAST (0.125–4 mg/L and 0.5–16 mg/L, respectively). MICs obtained by broth microdilutions of L-AmB were considerably higher than those of conventional AmB, regardless of the method (CLSI; 0.125–>16 mg/L, and EUCAST; 2–>16 mg/L). Contrary to CLSI and EUCAST (both 0.125–16 mg/L), Etest had a narrower MIC range for ISA (0.032–0.5 mg/L).

Table 2 presents the EA between broth microdilution and Etest results. AmB and ISA susceptibility results obtained by broth microdilution methods showed a good correlation within two dilutions (CLSI 98%, EUCAST 95.5%) but not for L-AmB (71.4%). In contrast, 90% EA was not reached for the Etest versus broth microdilution results. However, there were notable differences between the EA of Etest versus CLSI of AmB and ISA (79.6% and 77.6%) compared to the EA of Etest versus EUCAST of AmB and ISA (49.5% and 46.4%). The EA between EUCAST and CLSI results of AmB was excellent for both groups of *A. terreus* s.s. (100%) and non-s.s. (96.7%). According to the Etest versus CLSI comparison of AmB, *A. terreus* non-s.s. had a better EA (82.2%) than *A. terreus* s.s. (74%); however, both did not meet the acceptable range for EA (>90%). The agreement between Etest and EUCAST of AmB for *A. terreus* s.s. versus *A. terreus* non-s.s. did not reach 90%, in accordance with the agreement between all isolates. Concerning the EA of EUCAST versus CLSI of L-AmB, *A. terreus* non-s.s. showed a higher agreement (79%) than *A. terreus* s.s. (54%), but neither reached 90% of agreement. For ISA, EUCAST versus CLSI showed the best agreement (95%), followed by Etest versus CLSI (77.6%) and Etest versus EUCAST (46.4%).

## 4. Discussion

The main goal of AFST is to guide clinical decisions by providing reliable data, which is sometimes not possible, and some antifungal agents have different in vitro and in vivo responses [30]. Developing standardized AFST methods is crucial for predicting the response of fungal infections to a treatment and facilitating interlaboratory comparisons and agreement [19,31]. Broth microdilution AFST methods have been standardized by two organizations, CLSI and EUCAST. Several reports have discussed the differences between these two techniques; however, their results have been shown to be comparable and are used worldwide [32,33]. Mold susceptibility testing, using reference methods, albeit accurate, is less common in routine laboratories due to its labor-intensive and time-consuming procedures, especially in critical cases [19,34].

The commercial agar-based AFST methods, such as Etest, were found to be suitable alternatives to reference broth microdilution methods. These commercial methods resulted in faster turnaround times and simplified the inhibitory value evaluation process [19,35,36]. However, not all antifungal agents, such as L-AmB, are available (or approved) to be tested by these methods [35]. Moreover, some commercial techniques can provide susceptibility results that contradict those obtained using reference methods [37].

Different AFST methods are routinely used across countries or local laboratories, and their results can be affected by subtle variations (for example, glucose content, inoculum size, incubation time and temperature, and reading interpretation), which may explain the different MIC classifications and interpretations and, consequently, various epidemiological cutoff values (ECVs) and CBPs [38,39,40]. For instance, EUCAST does not recommend AmB for treatment of *A. terreus*, while Etest shows low AmB MICs against *A. terreus*. Therefore, it is crucial to know whether the results obtained by different methods are compatible, and how the heterogeneity of the results of the AFST methods and technical uncertainty affect the final interpretation. Hence, the present study evaluated the concordance between the reference methods (EUCAST and CLSI) and the commercial technique (Etest) while testing AmB, L-AmB, and ISA. Voriconazole is the first-line treatment for invasive aspergillosis; alternatively, ISA and L-AmB can be substituted [41,42].

The present study found that the level of agreement between methods varied depending on the antifungal agent and the tested method. Generally, the EUCAST method generated the highest MIC GM of AmB, L-AmB, and ISA, and the Etest method produced the lowest MIC GM for AmB, and ISA. The findings are consistent with a previous study that demonstrated that Etest for *Aspergillus* spp. usually provides lower MICs than standard microbroth dilution methods, CLSI, or EUCAST [43]. Consistent with another study [44], L-AmB resulted in higher MIC GM than AmB, regardless of the method of tested broth microdilution. L-AmB’s relatively high MICs may be due to incomplete and variable release of the active form of the drug into the medium in vitro [45]. More detailed experiments are required to comprehend the dynamics of lipid-associated polyenes under in vitro test conditions.

In the absence of AmB CBPs, based on the ECVs of EUCAST (wild-type, ≤8 mg/L), 99.2% of isolates were categorized as wild-type [22]. According to CLSI ECVs (wild-type, 4 mg/L), 100% of isolates were classified as AmB wild-type [21]. There are no established AmB CBPs and ECVs for Etest, but based on the suggested ECVs of AmB by Dannaoui and Espinel-Ingroff (wild-type, 16 mg/L) [28], all tested isolates were wild-type. Considering AmB ECVs, all three tested methods yield almost the same results, indicating that ECVs do not fully represent the diverse MICs obtained by each method.

Based on the EUCAST CBPs (≤1 mg/L) and ECVs (wild-type, ≤1 mg/L) for ISA [22], 39.2% of isolates were resistant. Based on the CLSI ISA ECVs of (wild-type, 1 mg/L) [29], all tested isolates belong to the wild-type population. There are no established CBPs and ECVs of Etest for ISA. Considering the tested species separately, EUCAST obtained the highest ISA MIC GM for *A. iranicus* and *A. niveus*, followed by CLSI, which both disagreed with Etest. EUCAST and CLSI had the best EA for AmB and ISA, but Etest had no acceptable agreement with any of them, for which the Etest method demonstrated lower MICs. However, the EA of Etest and both broth microdilution methods did not reach the accepted range, but CLSI showed a higher EA with Etest than EUCAST. The EA comparison showed Etest could not be compared to EUCAST or CLSI when testing ISA and AmB’s in vitro activity against the *A. terreus* species complex. This study’s limitation is that L-AmB did not have gradient strips, so comparisons were not possible.

There were no significant differences between Etest and reference methods when *A. terreus* s.s. and *A. terreus* non-s.s. were compared separately. Nevertheless, susceptibility profiles of *A. terreus* s.s. showed some trends compared with those of *A. terreus* non-s.s. (Table 1). Considering the EUCAST method, MIC GM of AMB for *A. terreus* non-s.s. (2.73 mg/L) was higher than *A. terreus* s.s. (1.84 mg/L), with the highest MIC GM assigned to *A. hortai*, *A. citrinoterreus*, and *A. alabamensis*. There was no big difference between the overall MIC GM of the ISA of *A. terreus* s.s. (0.62 mg/L) and non-s.s (0.71 mg/L) by EUCAST; however, the MIC GM of some species, such as *A. iranicus* and *A. niveus,* showed a higher value than those of *A. terreus* s.s.

In agreement with previous studies [28,46], this study emphasizes the need to determine the Etest-based ECVs and CBPs for antifungal agents such as AmBs and ISA and not interpret Etest MIC results based on the ECVs and CBPs of EUCAST or CLSI. Furthermore, ECVs should not be used instead of breakpoints, even though they can facilitate MIC interpretation. Prospectively collecting and pooling MIC data could confirm the calculated Etest-based ECVs, especially for AmB and ISA against *A. terreus*. On the basis of the results, the present study recommends caution regarding utilizing the Etest method for the AFST of AmB and ISA against *Aspergillus* section *Terrei* isolates since it is unclear whether the lower MICs determined with the Etest might affect the interpretation and choice of treatment and, subsequently, the in vivo outcome.

## Figures and Tables

**Figure 1 jof-09-00306-f001:**
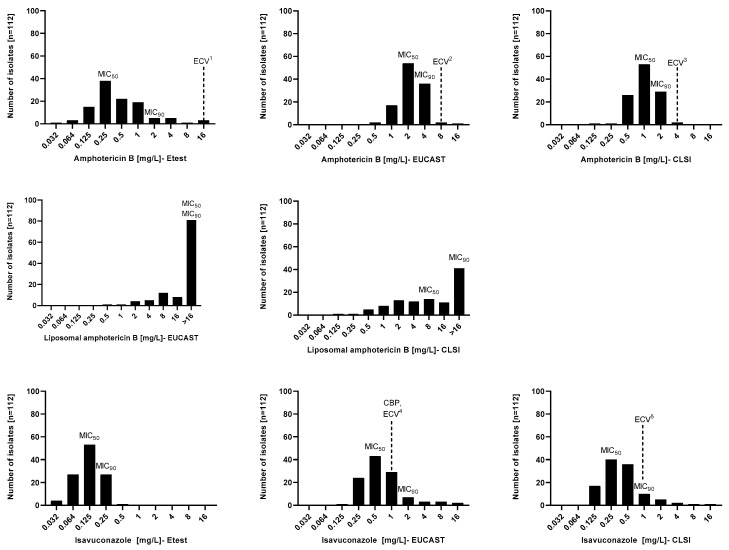
Minimal inhibitory concentration (MIC) distributions of amphotericin B, liposomal amphotericin B, and isavuconazole performed by EUCAST, CLSI, and Etest methods against *Aspergillus* section *Terrei*. 1. [28], 2. EUCAST database (https://www.eucast.org/mic_and_zone_distributions_and_ecoffs (accessed on 18 January 2022)), 3. [29], 4. EUCAST database (https://www.eucast.org/mic_and_zone_distributions_and_ecoffs (accessed on 18 January 2022)), (https://www.eucast.org/astoffungi/clinicalbreakpointsforantifungals (accessed on 18 January 2022)), and 5. [29]. ECV, epidemiological cutoff value; CBP, clinical breakpoint.

**Table 1 jof-09-00306-t001:** Susceptibility profiles of amphotericin B (AmB), liposomal amphotericin B (L-AmB), and isavuconazole (ISA) against *Aspergillus* section *Terrei,* using EUCAST, CLSI, and Etest methodologies. MIC_50_/MIC_90_ (MICs inhibiting ≥50% and ≥90% of strains, respectively), range, and GM (geometric mean) values are presented for AmB, L-AmB, and ISA. MIC_50_ and MIC_90_ are only shown for species with 5 or more isolates.

Species (Number of Isolates)	Method	MIC [mg/L]											
		AmB				L-AmB				ISA			
		MIC_50_	MIC_90_	Range	GM	MIC_50_	MIC_90_	Range	GM	MIC_50_	MIC_90_	Range	GM
*Aspergillus terreus* (n = 50)	Etest	0.25	1	0.032–2	0.27	-	-	-	-	0.125	0.25	0.064–0.25	0.14
	EUCAST	2	4	0.5–4	1.84	>16	>16	0.5–>16	6.28	0.5	1	0.25–8	0.62
	CLSI	1	2	0.125–2	0.91	4	>16	0.125–>16	2.96	0.25	1	0.125–4	0.36
*Aspergillus hortai* (n = 11)	Etest	0.5	0.5	0.064–1	0.37	-	-	-	-	0.125	0.25	0.064–0.25	0.133
	EUCAST	4	4	2–4	3.31	>16	>16	8–>16	8	0.5	1	0.25–2	0.53
	CLSI	2	2	0.5–2	1.29	16	>16	2–>16	5.66	0.25	0.5	0.125–0.5	0.28
*Aspergillus citrinoterreus* (n = 34)	Etest	0.5	4	0.125–16	0.61	-	-	-	-	0.064	0.25	0.032–0.25	0.09
	EUCAST	2	4	1–16	2.89	>16	>16	2–>16	7.25	0.5	1	0.125–4	0.48
	CLSI	1	2	0.5–4	1.28	>16	>16	0.25–>16	2.59	0.25	0.5	0.125–2	0.32
*Aspergillus alabamensis* (n = 9)	Etest	2	12	0.5–16	2.64	-	-	-	.	0.064	0.125	0.064–0.125	0.09
	EUCAST	2	4	2–4	2.72	>16	>16	2–>16	2	1	2	0.5–2	0.93
	CLSI	1	1	0.5–2	1	8	>16	1–>16	6.56	0.5	1	0.25–1	0.43
*Aspergillus iranicus* (n = 5)	Etest	0.25	1	0.25–2	0.66	-	-	-	-	0.25	0.25	0.125–0.5	0.25
	EUCAST	1	2	1–4	1.52	>16	>16	2–>16	2	4	16	2–16	6.96
	CLSI	0.5	0.5	0.5–1	0.57	8	>16	1–>16	4	2	8	0.25–16	3.03
*Aspergillus niveus* (n = 2)	Etest	-	-	0.125–0.5	0.25	-	-	-	-	-	-	0.125–0.25	0.18
	EUCAST	-	-	1–2	1.41	-	-	8–>16	8	-	-	2–4	2.83
	CLSI	-	-	0.25–0.5	0.35	-	-	4–8	5.66	-	-	2	2
*Aspergillus neoafricanus* (n = 1)	Etest	-	-	0.25	-	-	-	-	-	-	-	0.25	-
	EUCAST	-	-	4	-	-	-	>16	-	-	-	1	-
	CLSI	-	-	1	-	-	-	>16	-	-	-	1	-
All isolates (n = 112)	Etest	0.25	2	0.032–16	0.45	-	-	-	-	0.125	0.25	0.032–0.5	0.12
	EUCAST	2	4	0.5–16	2.29	>16	>16	0.5–>16	6.12	0.5	2	0.125–16	0.67
	CLSI	1	2	0.125–4	1.01	8	>16	0.125–>16	3.39	0.25	1	0.125–16	0.39

**Table 2 jof-09-00306-t002:** Essential agreement between broth microdilution and Etest results of *Aspergillus* section *Terrei* isolates (n = 112).

Species	Antifungal Agents	Essential Agreement		
*Aspergillus terreus* s.s. (n = 50)		**Etest vs. CLSI**	**Etest vs. EUCAST**	**CLSI vs. EUCAST**
	AmB	74.0%	44.0%	100%
	L-AmB	-	-	54.0%
	ISA	62.0%	58.0%	96.0%
*Aspergillus terreus* non-s.s. (n = 62)	AmB	82.2%	54.8%	96.7%
	L-AmB	-	-	79.0%
	ISA	67.7%	54.1%	95.1%
All species (n = 112)	AmB	79.6%	49.5%	98.2%
	L-AmB	-	-	71.4%
	ISA	77.6%	46.4%	95.5%

## Data Availability

Not applicable.

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
