# Peer review of "Aspergillus Section Terrei and Antifungals: From Broth to Agar-Based Susceptibility Testing Methods"

_jof, 2023, doi:10.3390/jof9030306_

Round 1

Reviewer 1 Report

This is a brief report, and the results obtained could help guide clinical decisions in the case of IFIs due to Aspergillus.

Observations in the Materials and Methods section:

2.1 Which molecular technique was used to identify the Aspergillus isolates, please provide a bibliographic reference (line 71).

2.3 A bibliographical reference should be added in the section where the technique with Etest strips in 2% RPMI glucose agar medium is described (line 91).

2.3 The incubation conditions of the fungus are not described (culture medium, incubation time, and temperature) to adjust to 0.5 with the McFarland tube (line 91). Approximately 106 CFU/mL, how is this value interpreted when studying a filamentous fungus?

 2.4 “except for strains with slow or insufficient growth, which were allowed an  additional 24 h at room temperature “ (line 97)

Why were the plates incubated at room temperature for an additional 24 h? Couldn't it have been incubated at 37oC even if it was less time?

Observation in the Results section:

The different panels of Figure 1 could be arranged horizontally 3 by 3, those corresponding to each antifungal and each test or methodology to facilitate the comparison of the results.

Author Response

Response to Reviewer 1:

Observations in the Materials and Methods section:

2.1 Which molecular technique was used to identify the Aspergillus isolates, please provide a bibliographic reference (line 71).

 The references describing the used molecular techniques have been added (Lines 79, 80).

 2.3 A bibliographical reference should be added in the section where the technique with Etest strips in 2% RPMI glucose agar medium is described (line 91).

 The references have been added (Line 96).

 2.3 The incubation conditions of the fungus are not described (culture medium, incubation time, and temperature) to adjust to 0.5 with the McFarland tube (line 91). Approximately 106 CFU/mL, how is this value interpreted when studying a filamentous fungus?

 The incubation conditions have been added (Lines 91-96).

 The inoculum turbidity was adjusted to 0.5 McFarland standard (equivalent to 1×106), as has been described in the reference. The references have been added (Lines 96, 98).

 2.4 “except for strains with slow or insufficient growth, which were allowed an additional 24 h at room temperature “(line 97) Why were the plates incubated at room temperature for an additional 24 h? Couldn't it have been incubated at 37oC even if it was less time?

An additional 24 h of incubation at 37oC would result in overgrowth and make reading the results difficult, and less than 24h incubation at 37oC would not be comparable with the reality in routine laboratories.

This explanation has been added (Line 101).

Observation in the Results section:

The different panels of Figure 1 could be arranged horizontally 3 by 3, those corresponding to each antifungal and each test or methodology to facilitate the comparison of the results.

The arrangement of figure has been changed.

Reviewer 2 Report

In this paper, authors compared 3 antifungal susceptibility testing methods, in 112 Aspergillus section Terrei and 3 antifungals agents (amphotericin B, liposomal amphotericin B and isavuconazole). Two of these methods are reference methods and the third is widely used in clinical laboratory. Antifungal agents tested are used to treat invasive fungal infection. Thus, this study is instructive and provides data for routine antifungal susceptibility testing.

Comments:

Content of MOPS in agar plates is not specified (MOPS is recommended by manufacturer).

While incubation was 48H for broth microdilution methods, incubation for Etest method was 24H. Authors did not specified why incubation was different.

Interestingly, L-AmB and of AmB were tested. An explanation of the benefit of testing L-AmB would be appreciated.

Author Response

Response to Reviewer 2:

Comments:

Content of MOPS in agar plates is not specified (MOPS is recommended by manufacturer).

The contents of MOPS have been added (Lines 94, and 96).

 While incubation was 48H for broth microdilution methods, incubation for Etest method was 24H. Authors did not specified why incubation was different.

This explanation has been added (Line 101).

Interestingly, L-AmB and of AmB were tested. An explanation of the benefit of testing L-AmB would be appreciated.

An explanation for the rationale behind of selection of tested antifungal agents in the present study has been added in the introduction (Lines 49-54).

Reviewer 3 Report

In this study the authors present a direct comparison between the broth dilution assays standardized by EUCAST and CLSI with the use of commercial Etest strips in the determination of antifungal susceptibility testing of Aspergillus section Terrei strains for the antifungals Amphotericin-B and Isavuconazole. The results show a good agreement between the CLSI and EUCAST protocols, but not acceptable between either of them with Etest. This is an important consideration when using Etest to define the susceptibility profile of these species and highlight the need for further analyses before Etest can be deemed suitable for this purpose.

As mention below, I think the inclusion of liposomal AmB is important to demonstrate that the broth dilution assays do not work well, independently of the low agreement between EUCAST and CLSI. In mi opinion this should be highlighted.

SPECIFIC COMMENTS

-Line 44-45: it is not clear if the statement implies that AmB is the most common antifungal used to treat systemic fungal infections caused by any fungi, by Aspergillus spp or by A. terreus. This needs to be clarify, as the validity of the statement would not be the same in all cases.

- The introduction would benefit from a clearer explanation of the antifungals used to treat infections caused by A. terreus, linked to the selection of the antifungals used in this study.

- It would be good to define MIC90 and MIC50 the first time these terms are used.

-Liposomal AmB performs really bad by broth dilution CLSI and even worse for EUCAST. I think this result should be stated and discussed.

-In figure 1 legend, please define the abbreviations ECV and CBP.

-With the analysis displayed in Table 1, can the authors draw some conclusions about the susceptibility of sensu stricto versus cryptic species? Even if there was already knowledge about this, it would be interesting to highlight observed differences. For example, I see that A. iranicus seems more susceptible than A. terreus to Amb, but resistant to ISA. Was that known? Can the authors comment on this (and maybe other) observations or trends?

Author Response

Response to Reviewer 3:

SPECIFIC COMMENTS

-Line 44-45: it is not clear if the statement implies that AmB is the most common antifungal used to treat systemic fungal infections caused by any fungi, by Aspergillus spp or by A. terreus. This needs to be clarify, as the validity of the statement would not be the same in all cases.

The statement has been changed to be more clarified (Lines 44, and 45).

- The introduction would benefit from a clearer explanation of the antifungals used to treat infections caused by A. terreus, linked to the selection of the antifungals used in this study.

An explanation has been added to the introduction (Lines 49-54).

 - It would be good to define MIC90 and MIC50 the first time these terms are used.

The definition has been added (Lines 105-107).

 -Liposomal AmB performs really bad by broth dilution CLSI and even worse for EUCAST. I think this result should be stated and discussed.

It has been added in the result part (Lines 120-122), and discussion part (Lines 185-189).

 -In figure 1 legend, please define the abbreviations ECV and CBP.

It has been added to the Figure legend (Lines 146).

 -With the analysis displayed in Table 1, can the authors draw some conclusions about the susceptibility of sensu stricto versus cryptic species? Even if there was already knowledge about this, it would be interesting to highlight observed differences. For example, I see that A. iranicus seems more susceptible than A. terreus to Amb, but resistant to ISA. Was that known? Can the authors comment on this (and maybe other) observations or trends?

The mentioned point has been discussed (Lines 210-217).
